# TreePO: Enhancing Policy Efficacy and Inference Efficiency with Tree Modeling

## Abstract

Recent advancements in aligning large language models via reinforcement learning have achieved remarkable gains in solving complex reasoning problems, but at the cost of expensive on-policy rollouts and limited exploration of diverse reasoning paths. In this work, we introduce TreePO, involving a self-guided rollout algorithm that views sequence generation as a tree-structured searching process. Composed of dynamic tree sampling policy and fixed-length segment decoding, TreePO leverages local uncertainty to warrant additional branches. By amortizing computation across common prefixes and pruning low-value paths early, TreePO essentially reduces the per-update compute burden while preserving or enhancing exploration diversity. Key contributions include: (1) a segment-wise sampling algorithm that alleviates the KV cache burden through contiguous segments and spawns new branches along with an early-stop mechanism; (2) a tree-based segment-level advantage estimation that considers both global and local proximal policy optimization. and (3) analysis on the effectiveness of probability and quality-driven dynamic divergence and fallback strategy. We empirically validate the performance gain of TreePO on a set of reasoning benchmarks and the efficiency saving of GPU hours from 22% up to 43% of the sampling design for the trained models, meanwhile showing up to 40% reduction at trajectory-level and 35% at token-level sampling compute for the existing models. While offering a free lunch of inference efficiency, TreePO reveals a practical path toward scaling RL-based post-training with fewer samples and less compute. Codes and repo will be released.

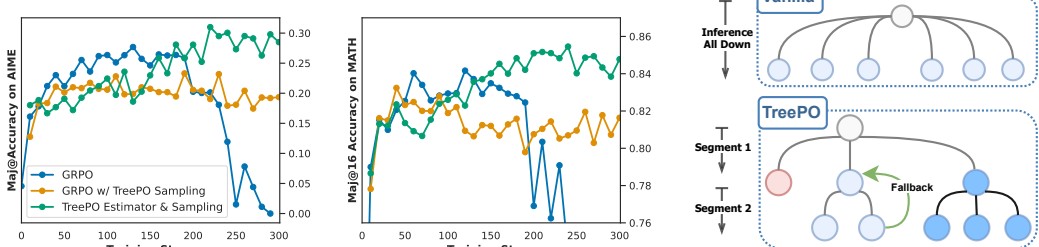

Figure 1: Demonstration of the Validation Performance Curves along Training based on Qwen2.5-7B (*Left, Mid*) and Demonstration of TreePO Sampling (*Right*). *Left, Mid*: Compared to the GRPO setting, although replacing additional treed-based sampling causes a slower convergence, it could stabilize the training.

## 1 Introduction

Reinforcement Learning (RL) has emerged as a powerful paradigm for enhancing the complex reasoning abilities of Large Language Models (LLMs) (Jaech et al., 2024; Shao et al., 2024; Yu et al., 2025). However, the efficacy and scalability of RL face fundamental constraints from two long-standing challenges: *exploration* (generating diverse responses) and *exploitation* (obtaining guidance from external feedback). In the context of LLMs, these challenges become even more pronounced, as models must generate sequences spanning thousands of tokens before receiving a single reward signal—which is typically sparse and delayed (Li et al., 2024; Guo et al., 2025a). This constraint creates two critical research challenges: (1) How can we enable LLMs to explore

potentially correct reasoning paths while maintaining or reducing computational costs? and (2) How can we accurately attribute sparse outcome rewards to the specific tokens that contributed to correct answers?

We present key observations that inspire our approach to addressing these challenges: standard RL approaches typically generate multiple *independent* trajectories for a single query—a strategy that is both computationally inefficient and conceptually sub-optimal. From a computational perspective, this approach creates paths with separate Key-Value (KV) caches, failing to utilize shared KV caching mechanisms that could significantly accelerate inference. Conceptually, continuing to explore paths already known to be impossible or incorrect, without early termination, represents a critical limitation in adaptability. That is, while this sampling strategy may appear simple to implement, its lack of structural design ultimately limits its effectiveness.

A promising sampling strategy is Monte Carlo Tree Search (MCTS) (Kocsis & Szepesvári, 2006) or its variants (Silver et al., 2017; Świechowski et al., 2023), which enables agents to leverage tree structures to achieve functions like early termination and roll back. Despite its promise, MCTS is often inefficient for LLM inference, requiring numerous sequential rollouts that are poorly suited for parallelized engines. Recent efforts have moved toward better utilization of LLM inference engines, recognizing that optimizing the data generation process itself is a critical frontier (Fan et al., 2025; Wang et al., 2025b). We believe this is the correct direction and accordingly propose a heuristic, self-guided, tree-based sampling mechanism designed to fully leverage the Key-Value (KV) cache mechanism. By structuring the rollout process as a tree, we maximize the reuse of shared prefixes as demonstrated in the Figure 1 (*Right*). Our findings show this approach can reduce 40% of trajectory-level inference time for the baselines on average (see §4.1), thereby improving computational efficiency without sacrificing performance.

To address the second question of credit assignment, our tree-based sampling structure naturally facilitates a more granular advantage estimation. This allows us to propose a new advantage function that is distinct from recent related works like TreeRL (Hou et al., 2025) and SPO (Guo et al., 2025b). While these methods also leverage tree or segment-based structures, their advantage calculations are primarily MCTS-like, focusing on the value difference between a parent and its child node to assign credit. Our approach, in contrast, models entire sub-trees as coherent sub-groups, enabling a more robust relative advantage calculation based on the collective outcomes of all descendants. More critically, our design is proven to be feasible for training directly from a base model, aligning with the "RL-zero" paradigm where reasoning capabilities are elicited without prior supervised fine-tuning (SFT). This stands in contrast to these related methods, which are demonstrated on models that have already undergone SFT.

In this paper, we introduce Tree-based Policy Optimization (TreePO), a framework that integrates these solutions into a unified RL pipeline. TreePO replaces inefficient independent rollouts with a computationally efficient and algorithmically flexible tree search. This structure not only improves sampling efficiency but also enables principled credit assignment and controllable exploration. We introduce novel heuristic sampling strategies, including dynamic divergence and probability-based fallback, which strategically allocate the generation budget to explore more diverse and promising reasoning paths. This transforms the rollout phase into a transparent and controllable search process, providing a powerful tool for analyzing the training dynamics of RL models. In summary, our contributions are:

- We introduce TreePO, a novel RL training scheme that replaces standard i.i.d. sequential sampling with a heuristic tree-based rollout mechanism. By implementing heuristic-driven exploration strategies, including dynamic divergence and probability-based fallback, this mechanism enhances the model's ability to explore the reasoning space effectively while significantly improving computational efficiency by leveraging KV-caching.

- We propose a new tree-based advantage estimation function that enables more precise credit assignment and is uniquely suited for training LLMs from a base model, without requiring an initial instruction tuning stage.

- We demonstrate through extensive experiments that TreePO provides a superior trade-off between computational cost and model performance, establishing a more efficient and scalable frontier for training large reasoning models.

## 2 TREEPO: A TREE-BASED TRAINING SCHEME FOR POLICY OPTIMIZATION

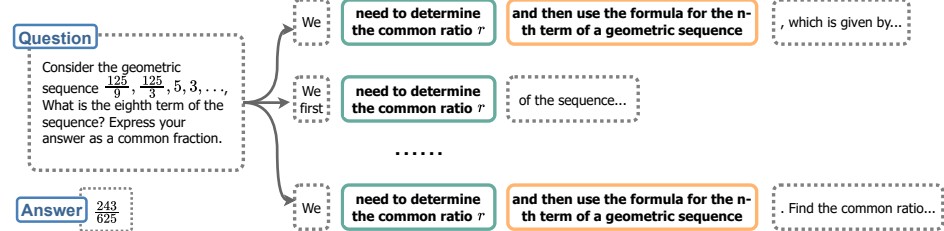

Figure 2: Multiple sampled trajectories from the same prompt, with shared reasoning segments highlighted in matching colors. Despite stochastic generation, key problem-solving steps are consistently reproduced.

### 2.1 CASE STUDY: THE ALIGNED MODEL PRODUCES SHARED PREFIX

We begin with an empirical observation on the structure of reasoning trajectories. Given a fixed prompt, we perform 16 independent stochastic rollouts using a temperature of $0.8$ to encourage diverse generation while preserving coherence. Upon close inspection, we find that despite the variation in final solutions, the generated trajectories share extensive overlapping segments, particularly in the early and intermediate stages of reasoning. As illustrated in Figure 2, components such as problem interpretation, variable assignment, and initial logical deductions appear nearly identical across multiple rollouts. These recurring segments are highlighted with consistent colors, visually demonstrating the emergence of stable reasoning prefixes.

This phenomenon indicates that, even under stochastic sampling, the model consistently follows a common path for the initial stages of reasoning before diverging at later decision points. Such redundancy across trajectories suggests a fundamental inefficiency in standard on-policy reinforcement learning: each rollout independently recomputes the same prefix tokens, leading to duplicated computation and KV cache storage. Since reasoning paths naturally form a tree-like structure where common prefixes branch into diverse continuations, it is both feasible and highly beneficial to model sequence generation as a tree-structured search process. By explicitly representing shared prefixes only once and amortizing computation over them, using TreePO avoids redundant forward passes. Furthermore, the natural branching points provide ideal locations for uncertainty-driven exploration, enabling efficient and targeted expansion of reasoning paths.

### 2.2 TREE-BASED ROLLOUT ALGORITHM

**Preliminaries.** For a given query $q_i \in Q$, we formalize the problem of complex reasoning with chain of thought (Wei et al., 2022) as the search algorithm to acquire a group of corresponding answers, $o_{i,j} \in O$, under a certain constraint of the computing budget. Specifically, we define the exact input of model as a prompt $p$, to distinguish the query itself as the input might contain additional context. In the TreePO sampling setting, we align terminology of RLVR and tree search to define:

    i. the query $q$ as the root node at depth $0$;
    ii. the number of complete trajectories as the tree width, $w$;
    iii. the maximum decoding steps of a trajectory as the depth, $d$;
    iv. the maximum decoding tokens of each time as the length of the segment, $l$; and
    v. branching budget $b$ for each segment node.

Under the context of RL training of large language models, the computing budget for sampling the trajectories of a given set of queries could be defined by the trajectory group size of each query (also noted as the tree width $w$), if the maximum trajectory length $d \times l$ is fixed.

**Segment-level Tree Sampling.** As shown in Figure 1 (*Upper Right*), the vanilla sampling design requires the model to conduct *token-level* decoding and stem multiple complete trajectories from the same query independently. We re-organize such a sampling progress into a hybrid of *segment-level* tree searching and *token-level* decoding as in Figure 1 (*Lower Right*): for each trajectory, the model generates a segment $s$ in max length $l$ step by step, until it hits the maximum response length or meets the *any self-designed criteria* of early stopping. We maintain a queue of prompts $P$ to manage the sampling progress, and assign the queries as the initial prompt set. For an input query set $q$, the

*token-level* decoding stops when the model generates `[EOS]` token or reaches the preset maximum segment token $\iota$; and the overall *segment-level* tree sampling progress ends when the prompt queue becomes empty ($P = \emptyset$). Specifically, given a $P$ in each step of decoding, the inference engine would produce a set of output segments in the exact number of $|P|$. And each generated segment will be either **appended** to existing contexts to form a new input prompt in the queue, or **stop generation** as a leaf node if it contains flawed sub-string patterns or answer `boxed`. We introduce the branching of each search paths by forking the corresponding prompts $b$ times before segment inference, where the value of $b$ is dynamically calculated and assigned by design (see the details in the following literature). To fulfill the requirement of acquiring $w$ trajectories for each $q$ when the searching paths stop early before the tree reaches $w$, we introduce the feedback mechanism and stem new branches from the stopped paths to achieve better efficiency. The pseudo code of the proposed sampling algorithm are described in Algo. 1, Appendix B.

**Branching and Fallback.** After reformulating the sampling progress into a tree-based search, a subtle balance between the rollout efficiency and model exploration space could be achieved by a well defined branching and fallback protocol. In TreePO, we define a vanilla $N$-ary tree as a baseline searching strategy, *i.e.*, the branching budget for a the root node $q$ (query) at depth $d$ is $N^d$ until it reaches the maximum width $w$. To avoid the inference loading skew caused by the scarce long responses and the over-bias on the short paths, we coordinate two balancing tree searching strategies with the inference engine:

- **Branching Budget Transfer**: As early stopped short search paths could derive a small request batch to the inference engine and thus cause low utilization, we assign the maximum branching budget $N^d$ at depth $d$ to all existing active paths evenly (or determined by heuristic information).
- **Depth-First Search Fallback**: To avoid sampling progress overly conducts fallback on the early stopped short paths and lose the capability of long complex reasoning, TreePO launches the fallback mechanism only when there is no active path for $q$ and the tree does not have enough trajectories $w_q < w$.

**Heuristic Sampling.** With the designed segment-level tree sampling protocol, we can now accordingly introduce a more fine-grained and flexible control over the sampling progress with heuristic information. Without waiting for external signals, the TreePO sampling could exploit more in the desired search space by leveraging heuristic control on early stopping, branching, and fallback strategies. We first introduce a simple early stopping trick for the flawed searching path by detecting the pattern with repetitive substrings within the new generated segment, which could reduce redundant computing, and forcedly prune the branches within the mumbling distribution that are usually generated by the less aligned base models. While conducting fallback, only those stopped paths containing formatted answer[1] or ending with `[EOS]` can be selected as the candidate to randomly fallback in segment level. Other than the average branching budget assignment and random fallback strategy, there are more possible customized heuristic metrics could be applied when maintaining efficiency of TreePO, as long as no additional bubble of the pipeline is introduced. In the later §**??**, we take advantage of the log probabilities to steer the sampling progress without additional cost, as they are calculated during token-level decoding and returned from the inference engines by default.

### 2.3 TREE-BASED ADVANTAGE ESTIMATION FOR POLICY OPTIMIZATION

We take the GRPO (Shao et al., 2024) optimizing objective and adopt the improved modifications proposed in DAPO (Yu et al., 2025) as our starting point, which further highlights clip-higher gradient, dynamic sampling, and token-level loss:

$$
\begin{aligned}
\mathcal{J}(\theta) = \quad & \mathbb{E}_{(q,a)\sim\mathcal{D}, \{o_i\}_{i=1}^{G}\sim\pi_{\theta_{\text{old}}}(\cdot|q)} \\
& \left[ \frac{1}{\sum_{i=1}^{G}|o_i|}\sum_{i=1}^{G}\sum_{t=1}^{|o_i|}\min\left(r_{i,t}(\theta)\hat{A}_{i,t}, \text{clip}\left(r_{i,t}(\theta), 1-\varepsilon_{\text{low}}, 1+\varepsilon_{\text{high}}\right)\hat{A}_{i,t}\right)\right],
\end{aligned}
\tag{1}
$$

$$
\text{s.t.} \quad 0 < \left|\{o_i \mid \texttt{is\_equivalent}(a, o_i)\}\right| < G.
$$

where

$$
r_{i,t}(\theta) = \frac{\pi_\theta(o_{i,t} \mid q, o_{i,<t})}{\pi_{\theta_{\text{old}}}(o_{i,t} \mid q, o_{i,<t})}, \quad \hat{A}_{i,t} = \frac{R_i - \text{mean}(\{R_i\}_{i=1}^{G})}{\text{std}(\{R_i\}_{i=1}^{G})}.
\tag{2}
$$

---

[1]In the context of math reasoning, we set this condition as including a legal answer surrounded by `boxed{}`.

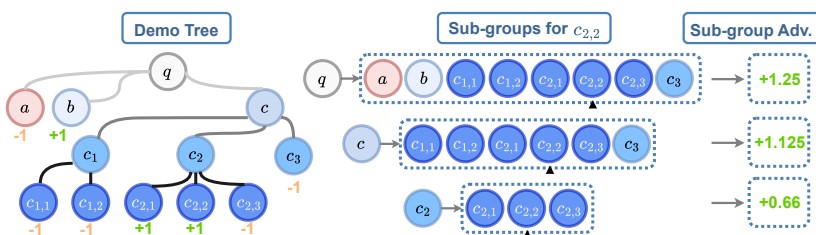

Figure 3: Demonstration of the TreePO Advantage Estimation. Assuming that the tree-based sampling has derived 8 trajectories (leaf) given a query $q$, we take node $c_{2,2}$ as an example to calculate the sub-group advantages. The tree-based sub-groups could be further defined by its predecessors $c_2$, $c$, and $q$. Thus the final advantages can be calculated as the averagely aggregated sub-group advantages.

Although the delicate modifications in DAPO (Yu et al., 2025) largely improve the stability of the vanilla GRPO, the parallel-generated responses could still look "homogeneous" in the sequence-level to the policy model under certain circumstances (e.g., inference with low temperature or train with an over-confident model). Benefiting from the tree structure in the proposed rollout algorithm, the searching paths could be sourced during advantage calculation. Given arbitrary trajectory $o_i$, it can be divided into multiple segments $s_j$ by its inference step $j$:

$$o_i = s_1 \oplus s_2 \oplus \cdots \oplus s_{j-1} \oplus s_j,$$
$$\{j \in J \mid j \le \mathrm{depth}_{\max}\} \tag{3}$$

Such a prior allow us to reveal the nuanced segment-level difference among the trajectories, and introduce more accurate intra-response variations for the advantages to alleviate the obscurity brought by similar responses. Leveraging the shared prefixes, the advantage estimation function for the trajectory could be further calibrated by the subgroups derived from the shared predecessor nodes for the leaf nodes. Let the root node $q$ be the sharing parent as the largest group $G$, we could denote a sub-group $G_j$ as the set of trajectories sharing the same predecessor node at inference depth $j$, satisfying:

$$G_{|J|} \subseteq G_{|J-1|} \subseteq \cdots \subseteq G_2 \subseteq G_1 \subseteq G,$$
$$\{j \in J \mid j < \mathrm{depth}_{\max}\} \tag{4}$$

Given the formulated sub-groups, we keep using the average reward within sub-groups as the advantage baselines and conduct mean pooling on the relative advantages as the aggregated estimation. Furthermore, we incorporate the global variance normalization strategy as in REINFORCE++ (Hu et al., 2025) to improve the robustness of the estimation function, as the probability-based branching could bring potential turbulent rollout rewards across queries, and conduct dynamic rejection sampling to remove the queries with all correct or all wrong responses as in DAPO (Yu et al., 2025). Hence the final TreePO advantage estimation function could be depicted as:

$$\hat{A}_{i,t} = \frac{\sum_{j=1}^{J} \hat{A}_{i,t,j}}{|J| \cdot std(\{\hat{A}_{i,t,j}\}^G)},$$
$$\hat{A}_{i,t,j} = R_i - \mathrm{mean}(\{R_{i,j}\}^{G_j}), \tag{5}$$
$$\text{s.t. } std(\{R_i\}^G) \ne 0$$

## 3 EXPERIMENT

### 3.1 SETTINGS

**Train and Eval.** The main experiments start from the `Qwen2.5-7B` base model (Qwen et al., 2025). To compare sampling methods on well-aligned LLMs, we also use `Qwen2.5-7B-Instruct` and `Qwen2.5-Math-7B-Instruct`. Training data are sourced from the MATH dataset Hendrycks et al. (2021) and the DeepScaler (Luo et al., 2025) collection. For evaluation, we use AIME 2024 (MAA, 2024), AMC 2023 (MAA, 2023), MATH500 (Hendrycks et al., 2021), MINERVA (Lewkowycz et al., 2022), and Olympiad Bench (He et al., 2024), with majority voting accuracy over 1000 samples as the main metric. The training was conducted on 64 GPUs, with a learning rate of $1e-6$ and a batch size of $512$ (more details in Appendix C).

**Tree Setting.** With a fixed response length, we search three sets of depth $d$ and segment token budget $l$: $\{28 \times 256, 14 \times 512, 7 \times 1024\}$. We set the total branching budget at *each depth* as $2^d$; specifically, the branching budget $b$ for each active path is $b = 2$, forming a binary tree search. The maximum tree width is set to $w = 16$, a parameter shared by the sequential sampling baselines. During training, we explore an enhancement to the initial branching budget **by randomly assigning 2 to 8 divergences**, aiming to improve diversity. We use "More Init Divergence" and "Fixed Init Divergence" to distinguish between these settings.

## 3.2 MAIN RESULTS

Table 1: Performance Comparison with Sequential Sampling with Maj@16 Acc.

| Model | AIME | AMC | MATH | MINERVA | Olympiad Bench | Overall |
|---|---|---|---|---|---|---|
| GRPO | 17.13% | 44.42% | 72.89% | 30.94% | 35.09% | 46.63% |
| GRPO w/ TreePO Sampling | 19.66% | 51.63% | 81.85% | 33.74% | 44.76% | 54.61% |
| TreePO w/ Fixed Init Divergence | **28.89%** | **56.63%** | 82.41% | **35.76%** | 47.75% | 56.88% |
| TreePO w/ More Init Divergence | 27.83% | 55.53% | **85.34%** | 34.98% | **49.15%** | **58.21%** |

Table 2: Performance Comparison Between Sequential and Tree-based Sampling with Maj@16 Acc.

| Model | Sampling | AIME | AMC | MATH | MINERVA | Olympiad Bench | Overall ↑ | GPU Hour ↓ |
|---|---|---|---|---|---|---|---|---|
| | Sequential | **28.89%** | 56.63% | 82.41% | 35.76% | 47.75% | 56.88% | 5.78 |
| TreePO w/ Fixed Init Divergence | 8x2048, $b = 2$ | 23.33% | **57.83%** | 81.80% | **36.76%** | 45.93% | 56.03% | 4.29 (↓26%) |
| | 8x2048, $b = 4$ | 23.33% | **57.83%** | 84.00% | 36.03% | 48.00% | 57.50% | 4.82 (↓17%) |
| | 8x2048, $b = 8$ | 26.67% | 55.42% | 83.60% | 36.40% | 46.22% | 56.60% | 5.09 (↓12%) |
| | Sequential | 27.83% | 55.53% | **85.34%** | 34.98% | **49.15%** | **58.21%** | 6.40 |
| TreePO w/ More Init Divergence | 8x2048, $b = 2$ | 21.52% | 53.99% | 81.89% | 33.93% | 44.41% | 54.67% | 3.65 (↓43%) |
| | 8x2048, $b = 4$ | 22.90% | 57.24% | 84.66% | 35.66% | 47.19% | 57.26% | 4.56 (↓29%) |
| | 8x2048, $b = 8$ | 26.21% | 56.72% | 85.23% | 35.02% | 48.81% | 58.06% | 5.05 (↓22%) |

The results of the main experiment set are provided in Table 1, where we use sequential sampling to validate the full potential of the model performances. Based on the provided results and the training curves in Figure 1 (*Left, Mid*), the introduction of tree-based methods — TreePO sampling and advantage estimator—serves to significantly enhance training stability and computational efficiency, albeit with a trade-off against raw convergence speed and peak accuracy in some configurations.

The effect of Tree Sampling is twofold. First, as shown in Table 1, adding TreePO sampling to the baseline GRPO model provides a substantial performance boost across all datasets, increasing the overall accuracy from 46.63% to 54.61%. This improvement is corroborated by the validation metric curves, where GRPO w/ TreePO Sampling (orange line) demonstrates far greater training stability compared to the volatile performance of the GRPO (blue line). Second, Table 2 reveals that while tree-based sampling does not always outperform a strong sequential baseline in final accuracy (e.g., 58.21% for Sequential vs. 58.06% for TreePO $b$=8 in the "More Init Divergence" model), it consistently and significantly reduces computation time, cutting GPU hours by 12% to 43%.

Beyond that, the TreePO advantage estimator, when used in conjunction with tree sampling, further enhances the training process, either with "More Init Divergence" (3.6% ↑) or "Fixed Init Divergence" setting (2.27% ↑). The green line in the validation curves shows the most stable and consistently high-performing trajectory during training. This indicates that the estimator component provides a more precise reward signal based on the tree hierarchy, guiding the model, and leading to more reliable convergence.

## 4 DISCUSSION

This section is organized around a set of research questions (RQs) that guide our investigation, with targeted ablation studies presented in the subsections. We start with a detailed analysis of offline sampling efficiency improvement to probe the basis of TreePO. We then provide a thorough study on the designed advantage estimation function and the tree sampling settings. We also place extended observations and discussions in Appendix D: sampling efficiency analysis, probability-based heuristic branching, and the performance curves along the scaling compute.

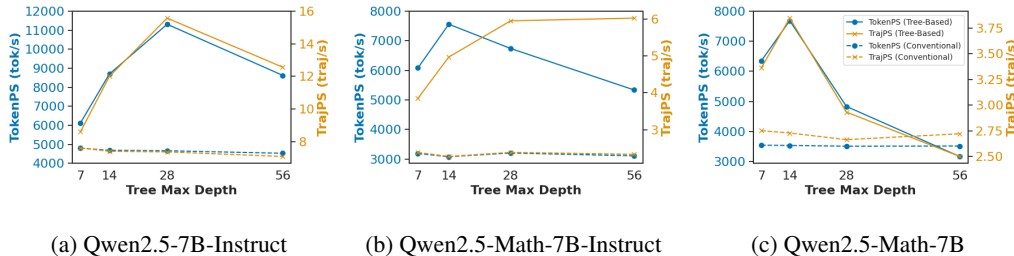

(a) Qwen2.5-7B-Instruct      (b) Qwen2.5-Math-7B-Instruct      (c) Qwen2.5-Math-7B

Figure 4: Performance comparison between Tree-based Sampling and Conventional Sampling across different tree depths.

## 4.1 SAMPLING EFFICIENCY ANALYSIS

**RQ.** Does tree-based sampling improve sampling efficiency relative to non–tree baselines, and under which segment and branching configurations?

**Setup.** To isolate efficiency, we conduct offline efficiency analyses using three variants of the Qwen2.5: Qwen2.5-Math-7B, Qwen2.5-Math-7B-instruct, and Qwen2.5-7B-instruct. We benchmark throughput on randomly sampled prompts from a held-out pool independent of model training. The experiments are carried out on H100 GPUs without any parallelization, maintaining a GPU utilization of 60%. Single inference maximum output length is determined by the tree segment length. Unless noted, each run processes a batch of **64 prompts** and, for tree-based sampling, **64 rollouts** per prompt. We fix a per-trajectory token budget $B = 7{,}000$ and vary *tree depth* $d$ and *max segment length* $L_{\text{seg}}$ subject to $d \times L_{\text{seg}} = w$ (segments are equal-length chunks). The non–tree baseline generates the same number of completions per prompt with identical sampling hyper parameters and the same budget $w$. We report *Tokens per second* (TokenPS; total processed tokens, including prefill and decode) and *Trajectories per second* (TrajPS; completions per second), measured as wall-clock throughput.

**Tree-based sampling generally improves efficiency.** Under the same batch size, rollout count, and budget $B$, tree-based sampling yields on average *+40% TrajPS* and *+30% TokenPS* across the three models (geometric mean across configurations). Figure 4 shows that both TokenPS and TrajPS peak at intermediate depth–segment combinations rather than grow monotonically with depth. The derived conclusions are intuitive: longer segments and shallower trees fit the prefilling stage better, which reduces repeated KV cache and attention computation; the decoding stage prefers deeper trees with more branches and parallel rollouts, better exploiting speculative execution and batched sampling. We also observe that if segments are too short, the extra recomputation offsets the gains from depth, and the peak appears where these opposing effects balance under the same trajectory budget. More observations regarding model priors and rollout scaling are discussed in Appendix D.1.

## 4.2 ANALYSIS ON THE TREEPO ADVANTAGE ESTIMATION

**RQ.** How does the design of the advantage estimation (e.g., subgroup aggregation choices) shape the optimization dynamics during training?

**Setup.** (1)–(3) use depth×segment $14 \times 512$ with a $512$-token fallback; (4) uses $7 \times 1024$ rollout but still a $512$-token fallback, inducing prefix misalignment. Figure 5 reports MATH/AIME accuracy, entropy loss, and response length, and the "Subgroup-size Weighted" curve serves as a reference baseline for comparison across variants.

$$\hat{A}_{i,t} = \frac{\sum_{j=1}^{J} |G_j| \cdot \hat{A}_{i,t,j}}{std(\{\hat{A}_{i,t,j}\}^{J-1}) \sum_{j=1}^{J} |G_j|}, \tag{6}$$

**Simple averaging across subgroups is better than subgroup-size weighting.** We further propose a modified estimation function Equation 6 from Equation 5 to validate whether a simple modification on the aggregation, based on subgroup size, is more appropriate for modeling the advantages. Averaging tracks higher accuracy on both MATH and AIME, with lower and more stable entropy and no unnecessary growth in response length. Size-weighting over-emphasizes large/easy subgroups and down-weights informative small/hard ones, whereas simple averaging preserves a balanced signal; we therefore adopt averaging in the method and keep size-weighting only for discussion.

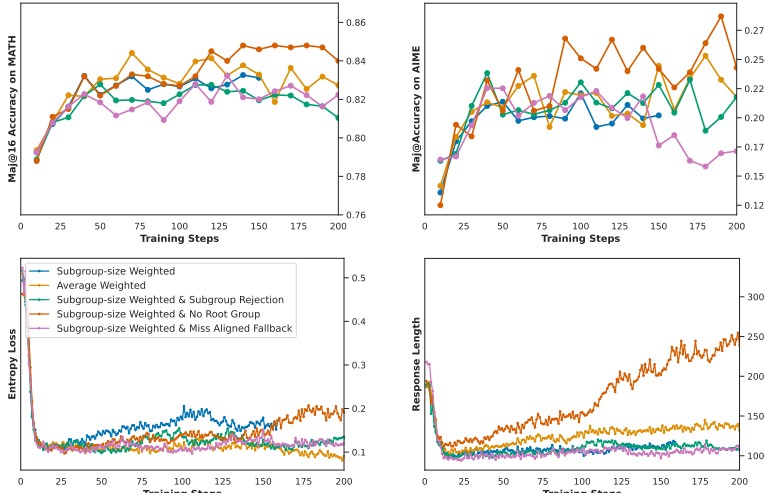

Figure 5: Study on the Terms in TreePO Advantage. These group of experiments sets the depth×segment as $7 \times 1024$ and uses the subgroup size weighted aggregation advantage as the baseline.

$$\hat{A}_{i,t} = \frac{\sum_{j=1}^{J} |G_j| \cdot \hat{A}_{i,t,j}}{std(\{\hat{A}_{i,t,j}\}^{J-1}) \sum_{j=1}^{J} |G_j|},$$
$$\hat{A}_{i,t,j} = R_i - mean(\{R_{i,j}\}^{G_j}),$$
$$s.t. \ \ std(\{R_{i,j}\}^{G_j}) \neq 0 \tag{7}$$

**Naïve subgroup rejection has marginal effect.** As shown in Equation 7, we tested subgroup-level dynamic rejection sampling (with extra subgroup hierarchy info). This DAPO-style rejection discards all-positive/all-negative subgroups to bias feedback signals—we expected better calibration, but accuracy dropped. Thus, in current settings, removing extremes doesn't always eliminate unwanted cases; we omitted subgroup-level rejection for final design simplicity.

**Removing the root-group advantage does not harm performance.** Using only aggregated subgroup advantages (dropping the root-group term) yields comparable curves, showing subgroup signals can approximate full-group optimization signals. This redundancy means the root term is non-essential, pointing to a promising direction for credit assignment analysis. This aligns with the fact that average weighting outperforms subgroup weighting—meaningful comparisons from *smaller subgroups* offer more effective calibration signals.

**Misaligned fallback degrades accuracy and inflates response length.** With $7 \times 1024$ segments but a $512$-token random fallback, trajectories can share an abstract tree prefix while being token-misaligned. Figure 5 shows a drop in AIME accuracy and a sharp rise in response length for the misaligned variant, highlighting that token-aligned segments are important for stable optimization and precise stopping behavior.

### 4.3 ANALYSIS OF EFFECTS OF SEGMENT BUDGET

**RQ.** How do the tree-sampling hyper parameters affect the convergence?

**Setup** We adopt the same subgroup-size weighted setting as in §4.2 to explore a the combination parameter of $d \times L_{seg}$. Here we set the $L_{seg} \in 128, 256, 512, 1024$ and adjust the maximum depth to fit the response length limit $7 \times 1024$ accordingly. The training curves are shown in Figure 6.

**Depth–segment trade-off, $14 \times 512$ is the sweet spot while $7 \times 1024$ underperforms.** Under the group-size weighted advantage, $14 \times 512$ attains the highest final MATH/AIME accuracy; $56 \times 128$ and $28 \times 256$ are close, whereas $7 \times 1024$ lags—especially on AIME—indicating that deeper trees with moderate segments provide stronger credit assignment than shallow rollouts with very long segments.

**Accuracy–length coupling, Better accuracy comes with longer generations.** The best-performing $14 \times 512$ also drives the largest growth in response length (and higher entropy), while

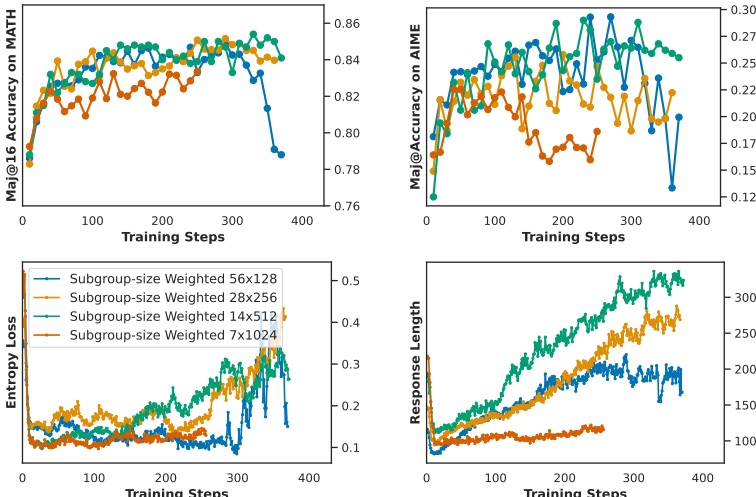

Figure 6: Study on the Online Depth-Segment under Setting the Group Size Weighted TreePO Advantage.

$7 \times 1024$ keeps outputs shorter but sacrifices accuracy. This suggests online TreePO benefits from more exploratory, longer reasoning traces; shorter traces trade accuracy for brevity.

## 5 RELATED WORK

**Efficient Sampling.** Recent work on efficient sampling for RL and inference concentrates on making the rollout loop lighter by batching many completions together, re-using the prompt KV-cache and hiding latency behind parallel decoding; typical examples are (Wu et al., 2025) (Hooper et al., 2025), (Zheng et al., 2025b), and (Guldogan et al., 2024), which all treat a prompt as a mini-batch and schedule tokens in groups so that GPUs stay busy. (Wang et al., 2025b) keeps this idea but breaks a large group into small "micro" groups, runs them with continuous inter-leaving, and adds a length-aware scheduler; this saves memory and keeps the buffer fixed, yet it does not look at the partial trajectories while they are generated, introduces extra scheduling logic, and leaves the advantage estimator untouched (Wang et al., 2025b). (Fan et al., 2025) improves wall-time by cutting every sampled chain after a short window and back-propagating early; the price is that long-range information is lost and credit assignment becomes harder.

**Segment-level Modeling.** A second line of research studies reinforcement learning with tree search. Recent systems such as (Wang et al., 2025a), (Hooper et al., 2025), (Hou et al., 2025) and (Guo et al., 2025b) build explicit trees and use them to explore many reasoning branches in one rollout, giving denser feedback than plain chain sampling (Wang et al., 2025a; Hooper et al., 2025; Hou et al., 2025; Guo et al., 2025b). TreeRL couples on-policy tree expansion with process-level rewards, but its trees stay shallow and the algorithm rolls one full answer to compute log-probabilities before it can branch again, which doubles the running time (Hou et al., 2025). (Lou et al., 2024) adopts an unconstrained tree and a "progress advantage" similar to Monte-Carlo returns; while this brings a simple tree-based update, it lacks depth control and is not validated against a frozen base policy. Similarly, ARPO (Dong et al., 2025) apply a segment-level entropy-guided divergence strategy based on the finished tool call trajectories, analogical to the FR3E (Zheng et al., 2025a) in math domain.

## 6 CONCLUSION

In this work, we introduced TreePO, a reinforcement learning framework designed to address the computational inefficiency and exploration instability in training large language models for complex reasoning. By reformulating on-policy rollouts as a segment-based tree search and using a hierarchical advantage estimator, TreePO significantly reduces reasoning computational costs while improving training stability and maintaining strong performance. The efficiency and structural modeling of TreePO open promising avenues for scaling reinforcement learning to more complex, long-horizon tasks such as multi-turn dialogue, tool use, and multi-agent systems.

## ETHICS STATEMENT

This work trains and evaluates models solely on publicly available math and problem-solving corpora; no personal or sensitive data were used, and no human subjects research was conducted, so IRB approval was not required. We complied with dataset licenses and attribution requirements and removed content if licensing status was unclear. While the proposed tree-based sampling can generally improve efficiency, it could also be repurposed to optimize harmful generations; accordingly, any released artifacts will include usage restrictions, safety filters, and refusal policies, and are not intended for high-stakes or autonomous decision-making.

## REPRODUCIBILITY STATEMENT

To ensure the reproducibility of our research, we has released our source code on GitHub and pre-trained model checkpoints on HuggingFace. The links will be updated in the paper after the anonymous period. We also provides hyperparameters and training setup required to replicate our experiments. We are confident that these resources will enable the research community to verify our findings and build upon our work.

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

## A  THE USE OF LARGE LANGUAGE MODELS (LLMS)

During the preparation of this manuscript, we utilized a Large Language Model (LLM) as a general-purpose writing assistant. Its role was confined to proofreading for grammatical errors, correcting typos, and improving sentence structure for better readability. The LLM did not contribute to the core research ideas, experimental methodology, or the substantive content of the paper.

## B  EXTENDED METHODOLOGY DETAILS

---
**Algorithm 1** Tree-based Sampling

---
**Require:** An array of queries $Q = \{q_1, q_2, \ldots, q_n\}$
**Ensure:** Rollout responses $O$ that satisfy the budget requirement for all $q \in Q$.
 1: $P \leftarrow Q$                               ▷ Init inference prompts with queries
 2: $P \leftarrow \text{BRANCHING}(P)$              ▷ Fork the prompts with designed policy
 3: **while** $P \neq \emptyset$ **do**
 4:     $S \leftarrow \text{INFERENCE}(P)$          ▷ Inference one step
 5:     $P^{last} \leftarrow P$
 6:     $P \leftarrow \emptyset$                    ▷ Clean up the inference queue
 7:     **for** $s_k$ in $S$ **do**                 ▷ Iterate throuhg the generated segments
 8:         **if** $\text{FINISH}(s_k)$ **or** $\text{FAILEDNODE}(s_k)$ **then**
 9:             $O \leftarrow O \cup \{\{p_k^{last} \oplus s_k\}$       ▷ Build the full response for final output
10:         **else**
11:             $P \leftarrow P \cup \{p_k^{last} \oplus s_k\}$        ▷ Concatenate the segment as new prompt
12:         **end if**
13:     **end for**
14:     $P \leftarrow \text{BRANCHING}(P)$          ▷ Fork the prompts with designed policy
15:     $P \leftarrow \text{FALLBACK}(P, O)$        ▷ Do fallback for unsufficient outputs
16: **end while**
17: **return** $O$                                 ▷ Return the final outputs

---

## C  EXPERIMENT DETAILS

**Model.** The main part of the reinforcement training experiments are trained from the `Qwen2.5-7B` base model (Qwen et al., 2025). Moreover, to further probe on the efficiency performance of the tree-based sampling on well aligned LLMs, we use the `Qwen2.5-7B-Instruct` and `Qwen2.5-Math-7B-Instruct` to compare the vanilla sequential and the tree-based sampling.

**Data and Evaluation.** One source of the training samples is the MATH dataset Hendrycks et al. (2021), deriving about 8 thousands queries of difficulty level 3 to 5 from, same as the setting in SimpleRL (Zeng et al., 2025). Another part of the training set consists 40 thousands samples from the DeepScaler (Luo et al., 2025) collection. For evaluation, we use the AIME 2024 (MAA, 2024), AMC 2023 (MAA, 2023), MATH500(Hendrycks et al., 2021), MINERVA (Lewkowycz et al., 2022), and Olympiad Bench (He et al., 2024). During validation and testing, we set the rollout N as 16 and use the majority voting accuracy via 1000 times of sampling as the main metric. For the overall metric, we use the weighted average among the individual benchmarks bases on the sizes of the test sets.

**Tree Setting.** With the constraint of response length, we search three sets of the depth $d$ and segment token budget $l$ of tree sampling in online training: $\{28 \times 256, 14 \times 512, 7 \times 1024\}$. We set the fixed total branching budget at *each depth* as $2^d$, *i.e.*, the branching budget $b$ of each active path is set as $b = 2$. Note that this forms a binary tree search paths if no early stop happens. And the maximum tree width is set as $w = 16$, where the sequential sampling baselines share the same group size parameter. During training, we explore whether additional initial branching budget **by randomly assign 2 to 8 divergences**, which is expected to improve the distribution diversity and thus break through the upper bound. Correspondingly, we use "More Init Divergence" and "Fixed Init Divergence" in the following literature to distinguish the settings of additional and fixed initial branching budget.

**Training.** We filter out the prompts longer than 1024 tokens, and set the response length as $7 \times 1024$ in training. The trainings run on 64 GPUs with the VeRL framework (Sheng et al., 2024) on FSDP mode, and use `V0` inference engine of vLLM (Kwon et al., 2023) as the inference backend. The learning rate is set as $1e - 6$ with 10 warm up steps. And the training batch size are set as 512 with the limit of maximum 20 epoch. The checkpoint saving interval as 50 steps. As the dynamic sampling strategy from DAPO is adopted, queries of $3 \times$ bsz would be sent to sampling out a group of 16

trajectories, where $512$ queries with $std(\{R_i\}^G) \neq 0$ are randomly selected. When there is not sufficient queries to form a training batch, maximum two other additional samplings will be conducted, which could cause a less training steps due to the enumeration logic of the data loader.

# D    EXTENDED DISCUSSION

## D.1    SAMPLING EFFICIENCY ANALYSIS (EXTENDED)

**The optimal depth–segment configuration is model-specific.**    Qwen2.5-7B-Instruct peaks at depth 28, likely because instruction-following finds a mid-depth balance: segments are not too short (better batched prefilling and context retention) while depth still yields sufficient decoding parallelism. Qwen2.5-Math-7B peaks at depth 14; for compute-intensive math reasoning, longer segments at shallower depth reduce repeated KV-cache and attention recomputation, improving throughput under the fixed budget. Qwen2.5-Math-7B-Instruct splits—TokenPS peaks at 14, whereas TrajPS peaks at 28 and 56—consistent with deeper trees (which shorten segments under the 7k-token budget) lowering token-level throughput via recomputation and decoder overhead, but raising trajectory-level throughput by enabling more branching and parallel rollouts.

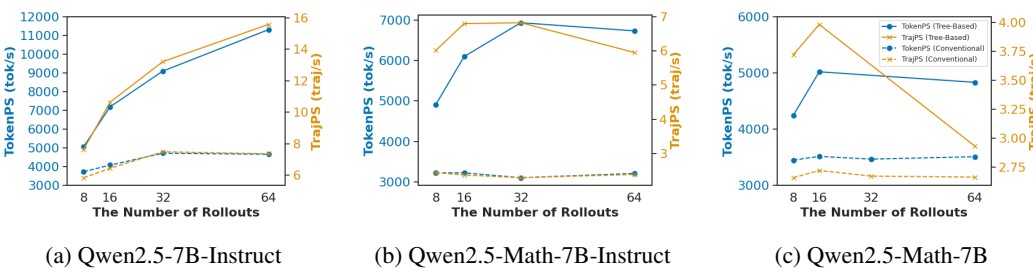

|                          |                                 |                      |
|:------------------------:|:-------------------------------:|:--------------------:|
| (a) Qwen2.5-7B-Instruct  | (b) Qwen2.5-Math-7B-Instruct    | (c) Qwen2.5-Math-7B  |

Figure 7: Performance comparison between Tree-based Sampling and Conventional Sampling across different numbers of rollouts.

**Rollout scaling is model- and workload-dependent.**    Qwen2.5-7B-Instruct shows nearly linear TokenPS/TrajPS growth as rollouts increase under tree-based sampling (with query count fixed at 64 and tree depth 28), reaching roughly $2\times$ the baseline thanks to shared-prefix *prefilling* and more parallel *decoding*; by contrast, standard autoregressive decoding yields only modest gains (Figure 7). Qwen2.5-Math-7B-Instruct maintains a stable $\approx 2\times$ speedup across rollout counts, as structured, semantically aligned math trajectories sustain high cache-hit rates and efficient KV reuse, keeping batched decoding effective. Qwen2.5-Math-7B is non-monotonic: throughput peaks around 16 rollouts, then TokenPS/TrajPS decline as trajectory divergence reduces shared prefixes, KV-cache fragmentation and management overhead grow, memory pressure rises, and batching efficiency degrades; the lack of instruction tuning further loosens output structure. Overall, more rollouts can boost parallelism and cache reuse but also amplify memory and synchronization costs when trajectories diverge, implying a model- and workload-dependent optimum.

## D.2    ANALYSIS OF PROBABILITY-BASED BRANCHING ASSIGNMENT

**RQ.**    How do different probability-based branching budget assignment strategies affect the convergence?

**Setup**    With segment-level control, TreePO sampling provide a more feasible environment to study the training dynamics of the LLM bounding to the decoding progress. Stemming from the TreePO "w/ More Init Divergence" setting, we conduct a set of experiments to control modify the branching assignment at a given depth $d$. Under this setting, the total branching budget $2^d$ is assign among the active paths conditioned on the log probabilities of their last segment, return from the inference engine. To prevent a sudden truncation of the search, the probabilities are passed through a softmax function with temperature set as $2.0$, and all the active search paths are guaranteed with at least one branching budget. The comparison between different branching budget controls are shown in Figure 8, where the "Low Prob Encourage" suggests the paths with lower probability get more branching budget, and vice versa for "High Prob Encourage". We also try with a more sophisticated "Low Prob Encourage" setting that the temperature of the softmax function is schedule from $5.0$ to $1.0$ across the training.

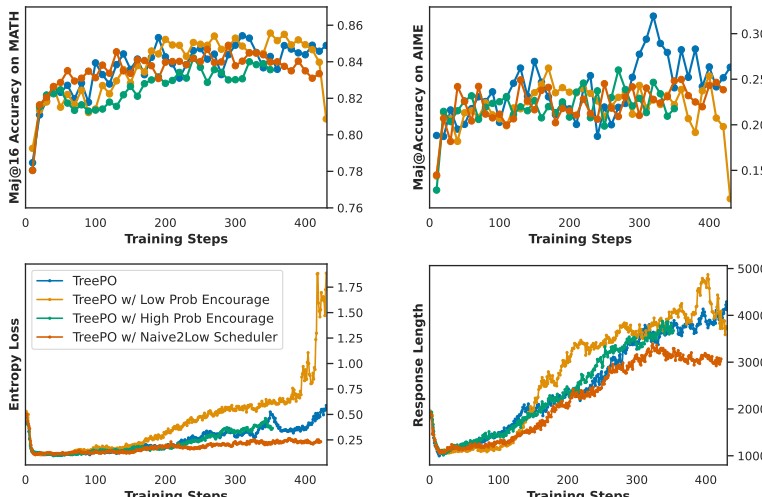

Figure 8: Study on a Probability-based Heuristic Tree Branching Budget Assignment.

**Monotonous Pattern Could Be Harmful.**  As shown in Figure 8, both static heuristic controls—"Low Prob Encourage" and "High Prob Encourage"—underperform the baseline and the scheduled variant. The "Low Prob Encourage" strategy, in particular, consistently yields the lowest accuracy on both benchmarks. This performance degradation is strongly correlated with a significant increase in response length and entropy loss, suggesting that forcing the model to explore low-probability states leads to less efficient and coherent search trajectories across the whole training. Conversely, the "High Prob Encourage" setting, while performing better, results in the lower entropy and shorter responses, indicating a potentially overly greedy search that may prune promising, less obvious paths too early. Even when the scheduled "Naive2Low" probability setting ensures a similar branching assignment scheme at the beginning of the training, it still does not provide significant advantages.

**Such Branching Control Does Not Show Significant Benefit Even with Higher Entropy.**  The most striking observation from our study is the disconnect between search diversity and task performance. The "Low Prob Encourage" setting was explicitly designed to increase exploration by allocating more resources to less likely search paths. This is reflected in its entropy loss, which is substantially higher than all other methods throughout training. However, this artificially inflated entropy does not translate into better results. Instead, it correlates with the worst performance on both benchmarks. This suggests that merely forcing the model to explore more diverse paths is not beneficial; the exploration must be meaningful. In this case, allocating budget to low-probability segments appears to push the model into irrelevant or erroneous reasoning paths, leading to longer, less effective solutions, as evidenced by the Response Length plot. The baseline maintains a more moderate entropy level, which proves more effective for complex reasoning tasks, indicating it strikes a better intrinsic balance between exploration and exploitation.

### D.3 COMPUTE SCALING FOR TREE SAMPLING

**RQ.**  How does the TreePO sampling performance scale along compute budget?

**Setup**  As the sampling mechanism has been modified, the scaling curve of a mono inference setting along compute does not necessarily follow the similar trend as sequential sampling. To investigate this, we analyze the test-time compute scaling of TreePO by evaluating weighted average model performances across benchmarks under various GPU hour budgets, as shown in Figure 9.

**Distinct Rules**  The experiment varies the tree divergence factor, which controls the number of branches generated at each divergence point, to observe its effect on the performance-compute trade-off. The results show that all configurations follow a predictable scaling pattern: performance improves with increased compute before eventually reaching a point of diminishing returns, which is consistent with established inference scaling observations. However, the key distinction from conventional sequential sampling becomes apparent when comparing the different divergence strategies. In sequential sampling, scaling compute is typically achieved by increasing the number of

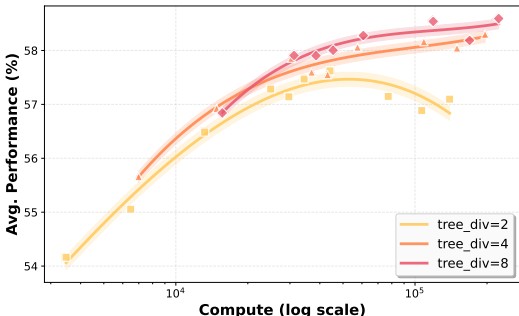

Figure 9: Test-time Compute Scaling of TreePO Sampling on the Aggregated Benchmark. The x-axis represents the compute budget on a log scale, while the y-axis shows average performance. Each curve corresponds to a different tree divergence factor $d = 2, 4, 8$. The results illustrate that a larger divergence factor can achieve higher peak performance at the cost of a larger compute budget, revealing a trade-off between the exploration strategy and computational cost that distinguishes it from the scaling behavior of conventional sequential sampling.

independent samples (N), which generally traces a single performance-compute curve. In contrast, TreePO generates a family of scaling curves, where each curve corresponds to a different internal search strategy controlled by $d$. At lower compute budgets, a smaller divergence factor ($d = 2$) is more efficient, achieving better performance for less cost. As the compute budget increases, wider search strategies ($d = 4$ and $d = 8$) become superior, with $d = 8$ ultimately reaching the highest peak performance. This demonstrates that the optimal TreePO sampling strategy is dependent on the available compute budget, allowing for more flexible "compute-optimal inference". Rather than simply scaling the number of samples, one can select the optimal tree structure to maximize performance for a given computational constraint.

