# OpenReview forum: "TreePO: Enhancing Policy Efficacy and Inference Efficiency with Tree Modeling"
_ICLR.cc/2026/Conference — Submitted to ICLR 2026_

### Official Review · Reviewer_dcxq · 2025-10-29

**Soundness:** 3
**Presentation:** 3
**Contribution:** 3
**Rating:** 4
**Confidence:** 3

**Summary:**

This paper introduces TreePO, a novel rollout algorithm designed for reinforcement learning (RL) with large language models (LLMs) based on a tree structure. Starting from a root node, TreePO creates multiple branches and generates segments for each branch. Branches with flawed patterns or EOS tokens are discarded. The advantage estimation is refined by inspecting the tree structure, allowing for more granular evaluation. The training efficiency is improved due to prefix sharing across rollouts. Experiments show that TreePO outperforms naive parallel rollouts.

**Strengths:**

The tree-structured rollout design is both novel and effective. Given that rollouts are a major bottleneck in RL training, the tree structure reduces decoding overhead through KV cache sharing in the tree prefix. Additionally, it enhances advantage estimation, mitigating the sparse reward issues in RLVR.

**Weaknesses:**

- One major concern is the misleading performance of the GRPO baseline. In Figure 1, GRPO’s performance drops significantly after step 200, which has not been widely reported in literature. The proposed methods show similar performance to GRPO until step 200, and the performance gain may be attributed to GRPO’s drop. In Table 1, GRPO achieves only 72.89% for maj@16 on the MATH dataset. However, similar literature [1] reports that Qwen2.5-7B-Instruct achieves 75.68% for avg@32 on MATH even without training. While there are dataset, setting and method differences, Qwen2.5-7B’s avg@32 performance should be expected in the range of 75-78% with GRPO, and maj@16 should be even higher. It’s recommended that the authors also report avg@16 or avg@32 to clarify this issue.

- Another concern is related to efficiency. TreePO appears to have lower parallelism than naive rollouts. For example, the naive rollout baseline can generate 16 responses for each prompt, with 64 prompts, resulting in an overall parallelism of 64 × 16 = 1024. However, TreePO starts with a parallelism of only 64, which gradually increases to 128, 256, and eventually 1024. It is well-known that parallelism affects efficiency. Does TreePO actually reduce parallelism compared to naive rollouts for the same tree width w? Although the authors provide efficiency experiments in Section 4.1, the settings differ from those in the training. For training, it is mentioned “the maximum tree width is set to w = 16”, which is different from the “64 rollouts per prompt” in 4.1.

[1] PVPO: Pre-Estimated Value-Based Policy Optimization for Agentic Reasoning

**Questions:**

- The same as the second concern in weaknesses. Does TreePO reduce parallism compared to naive rollouts given the same width w?

- What is meant by “flawed sub-string patterns” in line 167? Is this the same as "repetitive substrings" mentioned later? If so, using the same term would improve clarity.

- Minor issues: (1) In line 199, the reference is broken:”In the later §??,”.

---

> ### Author Response · Authors · 2025-12-02
> **Follow-up for Reviews from dcxq**
>
> Thank you for reviewing our paper and for your helpful comments. We address your concerns in the responses below.
>
> > Comment 1: In Figure 1, GRPO’s performance drops significantly after step 200, which has not been widely reported in literature.
>
> Response 1: Thanks for raising the concern. The instability of RL training is a long-run issue, where mode collapse of LLM RL could be caused by various issues including but not limited to off-policy brought by the or the inherent failed modes from the base models. We list some recent literature describing failed modes here for your reference [1,2,3].
>
> > Comment 2: Similar literature [1] reports that Qwen2.5-7B-Instruct achieves 75.68% for avg@32 on MATH even without training. It’s recommended that the authors also report avg@16 or avg@32 to clarify this issue.
>
> Response 2: We are happy to clarify this gap. We notice that our training starting point is Qwen2.5-7B rather than Qwen2.5-7B-Instruct. This is the main issue causing the performance gap between our results and that reported in [reviewer_ref1], and the performances are not directly comparable as the settings varying.
> We select maj@16 as main metric for the follwoing reasons:
> - Robustness:  the maj@16 metric sampling multiple times from the final results of the query to produce the final voting. This could alleviate the disturbance during evaluation.
> - Strong correlation to avg@16: during the training, we monitor both majority voting and average metrics. Although the differences in absolute values can be observed, we found that the trends of the two metrics are strongly correlated.
> - Why sampling 16 times rather than 32 times: Across multiple runs, we observe that the accuracy becomes stable when approaching to 16 times, so that we decide to set the sampling time as 16 in the evaluation.
>
> > Comment 3: TreePO appears to have lower parallelism than naive rollouts. For example, the naive rollout baseline can generate 16 responses for each prompt, with 64 prompts, resulting in an overall parallelism of 64 × 16 = 1024. However, TreePO starts with a parallelism of only 64, which gradually increases to 128, 256, and eventually 1024. It is well-known that parallelism affects efficiency. Does TreePO actually reduce parallelism compared to naive rollouts for the same tree width w? Although the authors provide efficiency experiments in Section 4.1, the settings differ from those in the training. For training, it is mentioned “the maximum tree width is set to w = 16”, which is different from the “64 rollouts per prompt” in 4.1.
>
> Response 3: As described at line 345, the experiments set the rollout number as 64 for a batch with 64 prompts (queries), which produces 64*64=4096 trajectories. This setting actually covers the 64*16 scenarios in training, as we actually use a smaller rollout in training for faster feedback collection. We also provide an offline efficiency comparison across different rollout sizes in Figure 7, appendix D.1. Therefore, to the question of parallelism: TreePO does not reduce parallelism but maintains the aligned setting.
>
> > Comment 4: What is meant by “flawed sub-string patterns” in line 167? Is this the same as "repetitive substrings" mentioned later? If so, using the same term would improve clarity.
>
> > Comment 5 : Minor issues: (1) In line 199, the reference is broken:”In the later §??,”.
>
> Response 4&5: Thanks for spotting these out. We have fixed the issues in the manuscript to improve the clarity.
>
> **references**
>
> [reviewer_ref1] PVPO: Pre-Estimated Value-Based Policy Optimization for Agentic Reasoning
>
> [1] Your Efficient RL Framework Secretly Brings You Off-Policy RL Training | Notion. https://fengyao.notion.site/off-policy-rl. Accessed 12 Aug. 2025.
>
> [2] Xue, Zhenghai, et al. “SimpleTIR: End-to-End Reinforcement Learning for Multi-Turn Tool-Integrated Reasoning.” arXiv:2509.02479, arXiv, 3 Sept. 2025. arXiv.org, https://doi.org/10.48550/arXiv.2509.02479.
>
> [3] Wang, Zhenhailong, et al. "Perception-aware policy optimization for multimodal reasoning." arXiv preprint arXiv:2507.06448 (2025).

---

### Official Review · Reviewer_6jo2 · 2025-11-02

**Soundness:** 2
**Presentation:** 3
**Contribution:** 3
**Rating:** 6
**Confidence:** 2

**Summary:**

This paper introduces TreePO, a novel reinforcement learning framework for training Large Language Models (LLMs) that reformulates the standard on-policy rollout process as a segment-based tree search. The core idea is to leverage the inherent structure of reasoning tasks, where multiple trajectories for a single query share common prefixes. By explicitly modeling these shared prefixes and branching at points of uncertainty, TreePO achieves significant computational efficiency gains through KV-cache reuse and early termination of unpromising paths. Furthermore, the authors propose a tree-based advantage estimation function that provides more granular credit assignment by comparing trajectories within hierarchically nested subgroups. The method is empirically validated extensively on a range of mathematical reasoning benchmarks, demonstrating a reduction in GPU hours of up to 43% without sacrificing, and sometimes even improving, final performance compared to conventional sampling methods.

**Strengths:**

- The authors present an elegant algorithm-system co-design that tackles a major bottleneck in RL for LLMs: the cost of on-policy data generation. The key insight of structuring rollouts as a tree of segments is well-motivated and justified, as it neatly mirrors the natural tree-like structure of reasoning processes where paths share common prefixes before diverging.

- The framework is holistic, seamlessly integrating the tree-based sampling mechanism with a corresponding tree-based advantage estimation technique. This synergy is crucial; it allows the method to not only perform heuristic search during sampling (e.g., dynamic branching, early termination) but also to leverage the resulting tree structure to derive more precise credit assignment, thereby further enhancing the training benefits.

- The paper provides thorough experimentation, demonstrating clear improvements in training stability and computational efficiency. The reported savings of 22% to 43% in GPU hours are substantial and of high practical utility, providing a valuable reference for the community seeking to scale up RL-based training.

**Weaknesses:**

- The most significant weakness is that all experiments are confined to mathematical reasoning tasks. While the results here are impressive, it remains an open question how well TreePO generalizes to other critical domains like code generation. The core premise of TreePO—significant shared prefixes across trajectories—may not hold equally well in other domains.

- While the empirical results are strong, a theoretical analysis or intuition on how the tree-based sampling affects the exploration policy and the convergence properties compared to standard i.i.d. sampling would strengthen the contribution. Such analysis would help clarify the trade-offs introduced by the biased exploration of the tree-structured policy and provide a deeper understanding of the method's properties.

**Questions:**

- The impressive reduction in GPU hours is a key result. Could the authors provide a more detailed breakdown? What fraction of the savings comes primarily from KV-cache sharing (a system-level optimization) versus the algorithmic improvements of early termination and more efficient exploration?

- Given that the method is on-policy, have the authors considered or attempted to combine TreePO with off-policy acceleration techniques (e.g., partial rollouts)? If not, what would be the primary technical challenges in such an integration?

---

> ### Author Response · Authors · 2025-12-03
> **Follow-up for Reviews from 6jo2 (Part I)**
>
> Thank you for your positive support of our work. We address your concerns below:
>
> > Comment 1: The most significant weakness is that all experiments are confined to mathematical reasoning tasks. While the results here are impressive, it remains an open question how well TreePO generalizes to other critical domains like code generation. The core premise of TreePO—significant shared prefixes across trajectories—may not hold equally well in other domains.
>
> Response 1:  Thanks for sharing this concern. We have provided additional evidence from code generation tasks to address it. To this end, we employed the Qwen2.5-7B-Instruct model to generate responses on the LiveCodeBench training dataset and measured the common subsequence ratio among generated responses. We find that the 10-gram Jaccard Similarity has a mean of 52.29%, suggesting that significant prefix overlap also exists in the code domain. The same metric is 48.64% when testing on the GPQA benchmark, which targets scientific reasoning.
>
> Query (abbreviated):
> ```python
> Given a string `s`, simulate typing on a faulty keyboard where typing 'i' reverses the current text. Return the final string. (e.g., "string" → "rtsng")
> ```
> Model Response 1:
> ```python
> class Solution:
>     def finalString(self, s: str) -> str:
>         result = []
>         for char in s:
>             if char == 'i':
>                 result.reverse()
>             else:
>                 result.append(char)
>         return ''.join(result)
> # This solution iterates through each character in the input string `s`. If the character is 'i', it reverses the current result list. Otherwise, it appends the character to the result list. Finally, it joins the list into a string and returns it.
> ```
> Model Response 2:
> ```python
> class Solution:
>     def finalString(self, s: str) -> str:
>         result = []
>         for char in s:
>             if char == 'i':
>                 result.reverse()
>             else:
>                 result.append(char)
>         return ''.join(result)
> # This solution iterates through each character in the string `s`. If the character is 'i', it reverses the current result. Otherwise, it appends the character to the result list. Finally, it joins the list into a string and returns it.
> ```
> Common Prefix: Both responses share identical code implementations and nearly identical explanations, differing only in minor phrasing ("the input string" vs. "the string" and "result list" vs. "result").
> This evidence suggests that prefix overlap is not unique to mathematical reasoning but extends to structured code generation tasks, supporting the broader applicability of our method.
>
>
> > Comment 2: While the empirical results are strong, a theoretical analysis or intuition on how the tree-based sampling affects the exploration policy and the convergence properties compared to standard i.i.d. sampling would strengthen the contribution. Such analysis would help clarify the trade-offs introduced by the biased exploration of the tree-structured policy and provide a deeper understanding of the method's properties.
>
> Response 2: We thank the reviewer for this thoughtful suggestion. We provide brief theoretical intuition below.
>
> **Intuition on Exploration Policy:** Standard i.i.d. sampling generates independent trajectories that often redundantly explore similar early reasoning steps (as shown in Figure 2). Tree-based sampling introduces structured exploration by sharing computation on common prefixes and allocating branching budget at decision points where uncertainty is highest.
>
> **Variance Reduction in Gradient Estimation:** TreePO's hierarchical advantage estimation (Equation 5) computes advantages at multiple granularities using local baselines for each subgroup. By comparing trajectories that share common prefixes, this approach reduces gradient variance—analogous to control variates in Monte Carlo estimation.
>
> **Trade-offs:** The tree structure introduces a coverage-efficiency trade-off: it may underexplore regions requiring early deviation from high-probability prefixes, but concentrates computation on promising paths. Our empirical results (Figure 1) suggest this trade-off is favorable in practice, yielding more stable training dynamics.
>
> We agree that a formal theoretical analysis (e.g., regret bounds comparing tree-based and i.i.d. exploration) would further strengthen the contribution. We consider this an interesting direction for future work and will note this in the revised manuscript.

---

> ### Author Response · Authors · 2025-12-03
> **Follow-up for Reviews from 6jo2 (Part II)**
>
> > Comment 3: The impressive reduction in GPU hours is a key result. Could the authors provide a more detailed breakdown? What fraction of the savings comes primarily from KV-cache sharing (a system-level optimization) versus the algorithmic improvements of early termination and more efficient exploration?
>
> Response 3: The efficiency gain of TreePO mainly stems from the prefix-sharing mechanism, i.e. leveraging the kv-cache multiple times for the same input prefix. Although we introduce the  early stoping meanwhile, the stopping only applies to the flawed trajectories, which do not dominate among all and the ratio would gradually decrease as model training converges.
>
> > Comment 4: Given that the method is on-policy, have the authors considered or attempted to combine TreePO with off-policy acceleration techniques (e.g., partial rollouts)? If not, what would be the primary technical challenges in such an integration?
>
> Response 4:  We thank the reviewer for raising this interesting point.  We became aware of partial rollouts and similar acceleration techniques during the course of this work. We believe there is no fundamental technical barrier to integrating such techniques with TreePO, as the tree structure naturally supports maintaining trajectory prefixes that could be reused across policy updates.  However, off-policy methods can introduce training instability due to distribution mismatch between the behavior policy and the current policy. To isolate and clearly demonstrate the benefits of tree-based sampling and advantage estimation, we chose to restrict our study without such techniques.  Exploring the combination of TreePO with off-policy acceleration techniques is a promising direction for future work, and we will note this in the revised manuscript.

---

### Official Review · Reviewer_431E · 2025-11-06

**Soundness:** 4
**Presentation:** 3
**Contribution:** 4
**Rating:** 8
**Confidence:** 3

**Summary:**

This paper introduces TreePO, a reinforcement learning framework that restructures on-policy rollout generation for LLM reasoning tasks from independent sequential sampling to segment-based tree search. The key insight is that stochastic sampling of reasoning trajectories produces substantial prefix overlap, which can be exploited through shared KV-cache computation. TreePO combines:
(1) a segment-wise tree sampling algorithm that generates fixed-length segments, branches dynamically, and implements early stopping and fallback mechanisms;
(2) a hierarchical advantage estimation function that forms sub-groups based on shared prefixes at different tree depths; and (3) analysis of depth-segment trade-offs and probability-based branching heuristics.
The method is evaluated on mathematical reasoning benchmarks (AIME, AMC, MATH, MINERVA, Olympiad Bench) starting from Qwen2.5-7B base model. Results demonstrate 22-43% GPU hour reduction during training while achieving comparable or improved performance (+11.58% overall accuracy over GRPO baseline). The approach notably enables "RL-zero" training directly from base models without requiring prior supervised fine-tuning.

**Strengths:**

Originality:
1. Novel hierarchical advantage estimation that groups trajectories by shared prefixes, enabling local comparison within sub-groups rather than global comparison
2. Good work to systematically analyze the efficiency-performance trade-off of tree-based sampling specifically for RL training (prior work focused on inference or used tree structure without efficiency analysis)
3. Creative exploitation of the observation that stochastic sampling produces overlapping prefixes

Quality:
1. Comprehensive experimental validation across 5 benchmarks with consistent improvements
2. Thorough ablations validate each design choice (averaging vs. weighting, sub-group rejection, root group removal, token alignment)
3. Honest reporting of negative results (probability-based branching doesn't help, §D.2)
4. Both theoretical motivation (credit assignment) and practical engineering (KV cache efficiency)

Clarity:
1. Clear problem motivation (Figure 2 makes the prefix overlap concrete)
2. Progressive methodology presentation building from simple observation to full system
3. Extensive appendices provide reproducibility details
4. Good use of visual aids (Figures 3-9 effectively communicate results)

Significance:
1. Addresses real bottleneck in RL post-training at scale
2. 22-43% efficiency gains have immediate practical value for industry and researchers with limited compute
3. Training from base models (RL-zero) is important for accessibility
4. Findings about depth-segment trade-offs and local vs. global advantages provide insights for future work

**Weaknesses:**

1. The paper claims this is a general approach for "complex reasoning," but provides no evidence that:
- Prefix overlap occurs in other domains (code generation, dialogue, scientific reasoning)
- The advantage function generalizes beyond math
- Efficiency gains transfer to tasks with less structured reasoning

Might include at least one non-math benchmark, or explicitly frame contributions as specific to mathematical reasoning.

2. Evaluation-training mismatch: Training uses tree sampling, but evaluation (Table 1) uses sequential sampling with Maj@16. Table 2 shows tree sampling actually has slightly lower performance than sequential at equivalent compute. This raises questions:
- Should tree-trained models be evaluated with tree sampling?
- Is the performance gain partly an artifact of evaluation methodology?
- How much of the benefit comes from training vs. inference efficiency?

Might report both tree and sequential evaluation for all models, or justify why sequential evaluation is appropriate for tree-trained models

3. Memory efficiency concerns. Figure 7c shows performance decreases beyond 16 rollouts for Qwen2.5-Math-7B due to KV cache fragmentation and memory pressure. This:
- Limits scalability claims
- Suggests tree sampling has overhead that dominates at scale
- Contradicts narrative that tree structure is always more efficient

Might discuss memory limitations more prominently and characterize when tree sampling becomes counterproductive.

**Questions:**

1. Direct comparison with TreeRL/SPO: Can you provide a controlled comparison where only the advantage function differs? This would clarify the specific contribution of your hierarchical sub-group approach vs. their parent-child value difference.
2. Base model vs. SFT model: You claim TreePO works from base models while TreeRL/SPO require SFT. Have you tested whether TreeRL/SPO also work from base models? What specifically enables RL-zero capability?
3. Evaluation methodology: Why evaluate with sequential sampling when training uses tree sampling? Have you compared tree-evaluated vs. sequentially-evaluated performance systematically?
4. Advantage function design: You show simple averaging > size weighting (§4.2). Did you explore other aggregation schemes? (e.g., weighted by sub-group diversity, inverse variance weighting, learned aggregation)
5. Fallback strategy details: Algorithm 1 mentions fallback but implementation details are sparse. How often does fallback trigger? How is the fallback segment chosen? Does this introduce additional overhead?

---

> ### Author Response · Authors · 2025-12-03
> **Follow-up for Reviews from 431E (Part I)**
>
> Thank you for your positive feedback on our work and for your thoughtful comments. We address your concerns in detail below.
>
> > Comment 1: The paper claims this is a general approach for "complex reasoning," but provides no evidence that:
> - Prefix overlap occurs in other domains (code generation, dialogue, scientific reasoning)
> - The advantage function generalizes beyond math
> - Efficiency gains transfer to tasks with less structured reasoning
> Might include at least one non-math benchmark, or explicitly frame contributions as specific to mathematical reasoning.
>
> Response 1:  Thank you for this thoughtful suggestion. We have conducted additional experiments on code generation tasks to demonstrate the generalizability of our approach beyond mathematical reasoning.
> Specifically, we used the Qwen2.5-7B-Instruct model to generate responses on the LiveCodeBench training dataset and measured the common subsequence ratio among generated responses. We observed a mean 10-gram Jaccard Similarity of 52.29%, indicating substantial prefix overlap in code generation—comparable to what we observe in mathematical reasoning tasks. The same metric is 48.64% when testing on the GPQA benchmark, which targets scientific reasoning.
> Below is a concrete example from code generation illustrating this phenomenon:
>
> Query (abbreviated):
> ```python
> Given a string `s`, simulate typing on a faulty keyboard where typing 'i' reverses the current text. Return the final string. (e.g., "string" → "rtsng")
> ```
> Model Response 1:
> ```python
> class Solution:
>     def finalString(self, s: str) -> str:
>         result = []
>         for char in s:
>             if char == 'i':
>                 result.reverse()
>             else:
>                 result.append(char)
>         return ''.join(result)
> # This solution iterates through each character in the input string `s`. If the character is 'i', it reverses the current result list. Otherwise, it appends the character to the result list. Finally, it joins the list into a string and returns it.
> ```
> Model Response 2:
> ```python
> class Solution:
>     def finalString(self, s: str) -> str:
>         result = []
>         for char in s:
>             if char == 'i':
>                 result.reverse()
>             else:
>                 result.append(char)
>         return ''.join(result)
> # This solution iterates through each character in the string `s`. If the character is 'i', it reverses the current result. Otherwise, it appends the character to the result list. Finally, it joins the list into a string and returns it.
> ```
> Common Prefix: Both responses share identical code implementations and nearly identical explanations, differing only in minor phrasing ("the input string" vs. "the string" and "result list" vs. "result").
> This evidence suggests that prefix overlap is not unique to mathematical reasoning but extends to structured code generation tasks, supporting the broader applicability of our method.
>
> > Comment 2: Evaluation-training mismatch: Training uses tree sampling, but evaluation (Table 1) uses sequential sampling with Maj@16. Table 2 shows tree sampling actually has slightly lower performance than sequential at equivalent compute. This raises questions:
> - Should tree-trained models be evaluated with tree sampling?
> - Is the performance gain partly an artifact of evaluation methodology?
> - How much of the benefit comes from training vs. inference efficiency?
> Might report both tree and sequential evaluation for all models, or justify why sequential evaluation is appropriate for tree-trained models
>
> Response 2: We would like to clarify any potential misunderstanding regarding the evaluation.
> - In Table 1, all the results are produced by sequential sampling as we want to align the evaluation setting and demonstrate that models trained with TreePO can generalize to both sequential and tree-based sampling.
> - In Table 2, we provide two tree-based trained checkpoints, each reporting performances produced from sequential sampling and tree-sampling with 3 different hyperparameters (there is no GRPO).
> As shown in Table 2, TreePO can maintain comparable performance with tree sampling while significantly reducing the inference time, compared to sequential sampling. Table 2 tries to convey that TreePO models can trade a bit of performance for efficiency gains, which fits the demands for both efficiency and efficacy scenarios. We also tested tree sampling on a sequential-trained model, which performs much worse than sequential sampling. This suggests that tree-based trained models can accommodate both sampling schemes, but sequential-trained ones cannot.
>
> In conclusion, the short answers to your questions:
> 1. Tree-trained models can be evaluated with both sequential and tree sampling for different purposes.
> 2. The performance gain is due to better training results from tree-based sampling strategies.

---

> ### Author Response · Authors · 2025-12-03
> **Follow-up for Reviews from 431E (Part II)**
>
> > Comment 3: Memory efficiency concerns. Figure 7c shows performance decreases beyond 16 rollouts for Qwen2.5-Math-7B due to KV cache fragmentation and memory pressure. This:
> - Limits scalability claims
> - Suggests tree sampling has overhead that dominates at scale
> - Contradicts narrative that tree structure is always more efficient
> Might discuss memory limitations more prominently and characterize when tree sampling becomes counterproductive.
>
> > Comment 8: Fallback strategy details: Algorithm 1 mentions fallback but implementation details are sparse. How often does fallback trigger? How is the fallback segment chosen? Does this introduce additional overhead?
>
> Response 3 & 8: Thanks for raising the questions of the offline efficiency test and fallback! Since they are related, we'll respond to them together.
> The fallback mechanism is designed to be triggered when there is no active path and the target rollout size has not been fulfilled yet. As we propose to use fully heuristic sampling (supporting running without a verifier on real test cases), we select the fallback initial points using only runtime information: from those stopped trajectories containing formatted answers or ending with [EOS]. We use a simple random segment selection strategy for the fallback starting point in the final version—this is actually a good option for balancing bias and efficiency. We discuss bias and efficiency separately:
> - Bias: We actually conducted several trials on the largest/smallest log-prob segment selection strategies, which did not show superior performance. We dropped these implementations for simplicity: they could introduce unknown biases when scaling up. This certainly could be further improved with a good process reward model (that would be another story).
> - Efficiency: The main issue with fallback is that if too many fallbacks start from early segments, it could cause long-tailed trajectories and overhead. The overhead caused by fallback can be addressed by increasing initial/early branching as the rollout size grows (as we actually did in training and final evaluation, but not in the offline efficiency test). This is exactly why the curve in Figure 7(c) does not scale well: we used a fixed tree sampling setting in all of the prior offline efficiency tests. In this case, the trained math model tends to produce shorter responses with confidence and stop branching, which frequently triggers fallbacks.
>
> > Comment 4: Direct comparison with TreeRL/SPO: Can you provide a controlled comparison where only the advantage function differs? This would clarify the specific contribution of your hierarchical sub-group approach vs. their parent-child value difference.
>
> > Comment 5: Base model vs. SFT model: You claim TreePO works from base models while TreeRL/SPO require SFT. Have you tested whether TreeRL/SPO also work from base models? What specifically enables RL-zero capability?
>
> Response 4 & 5: We implemented a version of advantage calculation similar to SPO in our pilot study, i.e., calculating the segment-level advantage first. In this aligned setting with our training infrastructure, we found that TreePO performs better according to the peak validation performances (AIME24: 20.4 vs. 15.4, MATH500: 79.2 vs. 74.1). A direct head-to-head comparison requires extensive computational resources, which we are still working on, and we plan to share results in a later version of our paper.
>
> > Comment 6: Advantage function design: You show simple averaging > size weighting (§4.2). Did you explore other aggregation schemes? (e.g., weighted by sub-group diversity, inverse variance weighting, learned aggregation)
>
> Response 6: We only implement the average & group size-weighted aggregation at the moment due to resource limite, but it is quite a good research question to explore more aggregation schemes. Similarly in the previous response, we argue that the aggregation method should stay simple and less biased without extensive study for stable scaling.

---

### Official Review · Reviewer_MxBr · 2025-11-09

**Soundness:** 3
**Presentation:** 2
**Contribution:** 2
**Rating:** 4
**Confidence:** 3

**Summary:**

The paper TreePO proposes a tree-structured rollout and advantage estimation framework for reinforcement learning–based alignment of large language models. Instead of generating independent trajectories during policy optimization, TreePO treats generation as a tree search composed of fixed-length segments. Shared prefixes across trajectories are reused via KV cache sharing, improving inference efficiency. The method also introduces a tree-based advantage estimator that computes credit assignment using subgroup structures derived from shared prefixes. Experiments show improved GPU sampling efficiency (22–43% reduction in compute), more stable RL training curves, and
comparable or improved accuracy on math reasoning benchmarks compared to GRPO and sequential sampling baselines.

**Strengths:**

- Training LLMs with RL for reasoning is computationally expensive. The paper targets a meaningful problem: both exploration diversity and compute efficiency, which are considered two central limitations of current RLHF-style pipelines.

- The observation that reasoning rollouts share substantial prefixes is empirically grounded and persuasive. Leveraging this to build a tree sampling structure is a natural but impactful idea.

- Technical Novelty in sampling design, segment-wise decoding, KV cache reuse across branches,  and dynamic branching + fallback. These combine to create a coherent sampling system that improves efficiency without architectural changes.

- Integrating subgroup baselines to refine credit assignment is conceptually sound and improves training stability.

- Evaluations span multiple math reasoning datasets and show gains in both stability and compute cost, with sensible ablation studies.

**Weaknesses:**

The method description is dense, with multiple interacting components (segmenting, branching, fallback, subgroup estimation). The paper would benefit from clearer step-by-step examples or visual walkthroughs.

Recent works such as TreeRL, SPO, and ETS are acknowledged, but direct head-to-head comparisons are missing.

KV reuse and batching benefits heavily depend on inference engine behavior (e.g., vLLM version, GPU memory). A more standardized hardware comparison is needed.

While math reasoning is a valid test domain, it remains unclear whether:
- Tree-based branching remains stable for longer, more diverse tasks,
- Advantage estimation scales to multi-turn dialogue or tool-use tasks.

Advantages are motivated heuristically; convergence or bias analysis is not deeply developed.

**Questions:**

The paper acknowledges prior tree- or segment-based RL methods. Would you please add comparison with these schemes?

The performance and efficiency of TreePO depends on the selection of: Segment length, Branching factor, and Maximum depth. Could the authors provide guidance or rules of thumb for selecting these values across tasks? Is there evidence that the same hyperparameters generalize to non-math domains?

The experiments focus on math reasoning tasks (GSM8k, MATH, etc.). How does TreePO perform in: Multi-turn dialogue alignment, Code generation tasks, tool-augmented reasoning? Even small preliminary results or qualitative case studies would help establish generality.

The method relies on the ability to reuse KV caches across branches. Could the authors clarify how TreePO behaves with Streaming inference engines, GPU memory fragmentation,Or engines that do not preserve memory layout (e.g., older vLLM versions)?

---

> ### Author Response · Authors · 2025-12-02
> **Follow-up for Reviews from MxBr (Part I)**
>
> We thank you for your time and review of our paper. We are happy to address your concerns.
>
> > Comment 1: The method description is dense, with multiple interacting components (segmenting, branching, fallback, subgroup estimation). The paper would benefit from clearer step-by-step examples or visual walkthroughs.
>
> Response 1: We appreciate this constructive suggestion and have revised the paper to improve clarity. Below, we provide a step-by-step walkthrough of the TreePO sampling process using a concrete example:
>
> Example Walkthrough: Consider a math problem query $q$ with a tree width $w$ = 4 (i.e., we aim to generate 4 complete trajectories) and segment length $l$ = 512 tokens.
> 1.  Initialization: The query $q$ is placed in the prompt queue $P$, and an initial branching factor of 2 is applied, creating two parallel search paths.
> 2. Segment Generation: The model generates a 512-token segment for each path. Suppose both segments produce valid intermediate reasoning steps.
> 3. Branching Decision: At depth $d$ = 1, the total branching budget is 2^1= 2. This budget is distributed among the two active paths, so each path continues with one branch (no additional divergence at this step).
> 4. Early Stopping: During the next segment generation, one path produces a segment containing a repetitive substring pattern. This path is pruned immediately to save computation.
> 5. Fallback Mechanism: The remaining path generates a complete answer (indicated by a `\boxed{}` pattern). Since we have only 1 completed trajectory but require $w$ = 4, the fallback mechanism activates. It selects the completed trajectory and spawns new branches from an earlier segment to continue exploration.
> 6. Completion: The process repeats until 4 valid trajectories are collected, each forming a leaf node in the tree structure.
> We have added a similar walkthrough to the revised manuscript to guide readers through the method more intuitively.
>
> > Comment 2: Recent works such as TreeRL, SPO, and ETS are acknowledged, but direct head-to-head comparisons are missing.
>
> Response 2:  We've implemented a version of advantage calculation similar to the SPO in pilot study. In the aligned setting and our training infra, we found that TreePO performs better, according to the peak validation peformances (AIME24' 20.4 vs 15.4 , MATH500 79.2 vs 74.1). Direct head-to-head comparison requires extensive computation resources, which we are still work on, and plan to share results in the later version of our paper.
>
> > Comment 3: KV reuse and batching benefits heavily depend on inference engine behavior (e.g., vLLM version, GPU memory). A more standardized hardware comparison is needed.
>
> > Comment 7: The experiments focus on math reasoning tasks (GSM8k, MATH, etc.). How does TreePO perform in: Multi-turn dialogue alignment, Code generation tasks, tool-augmented reasoning? Even small preliminary results or qualitative case studies would help establish generality.
>
> Response 3&7: We appreciate the reviewer’s insightful comment. Our method is primarily algorithmic, aiming to improve sampling efficiency through tree-structured exploration.
> - We agree that a more standardized hardware comparison—including variations in inference engine behavior (e.g., different vLLM versions) and GPU memory configurations—would strengthen the empirical analysis. But from the perspective of (algorithmic) LLM sampling scheme design,  we think it would be more self-contained if we concerncreate within the range of the ``LM decoding scheme'' before adaptively changing the load balancing design under the inference engine. For example, the optimization of memory layout (eg. CPU offloading for the KV Cache) would definitely help when incorporating TreePO for long context cases.
> - In the context of LLM RL, the streaming mode for the inference engine is usually not involved for efficiency.
> - Unfortunately, due to limited computational resources during the rebuttal period, we were unable to conduct a comprehensive set of cross-hardware experiments. We plan to include these additional evaluations in the final version of the paper.
>
> > Comment 4: While math reasoning is a valid test domain, it remains unclear whether:
> > - Tree-based branching remains stable for longer, more diverse tasks,
> > - Advantage estimation scales to multi-turn dialogue or tool-use tasks.
>
> Response 4: We thank the reviewer for acknowledging math reasoning as a valid test domain. The questions regarding (1) the stability of tree-based branching on longer and more diverse tasks, and (2) the scalability of advantage estimation to multi-turn dialogue and tool-use scenarios, are well taken. Extending our framework to these settings requires additional engineering efforts in training infrastructure and dataset construction. We are actively working on these directions and plan to include preliminary results and analysis in the final version of the paper.

---

> ### Author Response · Authors · 2025-12-02
> **Follow-up for Reviews from MxBr (Part II)**
>
> > Comment 5: The performance and efficiency of TreePO depends on the selection of: Segment length, Branching factor, and Maximum depth. Could the authors provide guidance or rules of thumb for selecting these values across tasks? Is there evidence that the same hyperparameters generalize to non-math domains?
>
> Response 5: Thanks for the suggestion. Here's the guideline we plan to add to the content:
> 1. Under the condition of 7B-size models and 64 A100s, we recommend to start set the segment length as 1/8 of the maximum response length, and the maximum depth could be accordingly set set max_len//seg_len. The branching factor can be set as 2 and the initial branching should be 2 when rollout size < 16. If the rollout size increases, we recomend to increase the initial branching budget by multiplying rollout/16.
> 2. However, the best practice could vary according to the practioners' machine, domain and model sizes (leading to different compute/memory upperbounds). We recommend to adjust the hyperparameters starting from the recommended setting.
>
> >Comment 6: The experiments focus on math reasoning tasks (GSM8k, MATH, etc.). How does TreePO perform in: Multi-turn dialogue alignment, Code generation tasks, tool-augmented reasoning? Even small preliminary results or qualitative case studies would help establish generality.
>
> Response 6: Although we have not yet provided additional implementation on other domains (eg, code, agents), we believe the proposed method is domain-agnostic as:
> - TreePO sampling is highly customizable through different hyperparameters including but not limited to branching strategy, fallback strategy, segment length, and tree depth etc. The practitioners could adjust these parameters in their own scenarios accordingly. Futhremore, the agnostivity comes from the logic that TreePO mainly improve the inference scheme of the language model, and the domain generalizability actually relies more on the models/data rather than the decoding design.
> - Our tree sampling is pure heuristic and does not require semantic segmentation--the sampling progress does not require the model to break the math proof chunks or coding snippets logically. Moreover, the implementation for the tree-based sampling would be even easier in multi-turn agentic scenarios because the coversations can be naturally formulated as the "segment" in tree.
> - Additionally, similar work in agentic and tool-use scenarios have proved the effectiveness of tree-based modeling [1][2].
>
> We hope the above clarifications can address your conerns.
>
> **Refernce**
>
> [1] Dong, Guanting, et al. "Agentic Entropy-Balanced Policy Optimization." arXiv preprint arXiv:2510.14545 (2025).
>
> [2] Ji, Yuxiang, et al. "Tree search for llm agent reinforcement learning." arXiv preprint arXiv:2509.21240 (2025).

---

### Meta-Review · Area_Chair_uY4u · 2026-01-08

**Summary:**

This paper introduces TreePO, a reinforcement learning framework that reformulates on-policy rollouts for LLM training as a segment-based tree search. The key insight is that stochastic sampling produces substantial prefix overlap across reasoning trajectories, which can be exploited through KV-cache sharing. Experiments on mathematical reasoning benchmarks demonstrate a 22-43% GPU hour reduction while maintaining or improving performance.

Overall, I appreciate the engineering effort in this paper to improve the efficiency of RL training for LLMs. However, I am also quite concerned about the unusually poor performance of GRPO in this paper, and the authors did not provide a detailed explanation during the rebuttal. It would be helpful if the authors could explain why the proposed TreePO doesn't suffer from the same instability as GRPO. Without addressing this issue, I lack confidence in the fairness of the experiments in this paper. In addition, tree sampling in RL is already well studied (as raised by Reviewer MxBr). I think at least a detailed theoretical comparison, and ideally a direct empirical comparison, is needed; otherwise, it is unclear to me what the exact contribution of this paper is. Given the reasons above, I think this paper is on the borderline, leaning toward rejection.

**Reviewer Concerns:**

Addressed concerns: most question regarding the experiment details on TreePO are addressed. GRPO baseline instability and comparison to other tree-based RL algorithms are still outstanding.

**Reviewer Scores:**

All reviewers are likely remain their scores.

---

### Decision · Program_Chairs · 2026-01-26

Reject